# Topic Model-based Analysis for Gaze and Education

## Abstract

Multimodal learning analytics have become increasingly important in enabling a deeper understanding of teaching and learning in educational research. However, in comparison to other multimodal learning data, there is a limited understanding of the trends within gaze learning data. To address this challenge, this study aims to identify latent topics in gaze learning data through topic modeling and scientometric analysis. We analyzed the abstracts of 573 peer-reviewed and conference proceeding papers that used gaze learning data, written in English, and published between 2008 and February 2024. The findings are as follows. First, three main topics were identified through topic modeling analysis: (learning analytics, multimodal learning, and inclusive learning). Second, the scientometric analysis revealed the structure in which diverse clusters in cited references and institutions are connected around the emerging topics. Based on these findings, the study would provide insights into research directions in both educational research and applications using gaze learning data.

## 1 Introduction

In alignment with an increasingly acknowledged emphasis on digital technology generating a vast amount of multimodal learning data within educational research, the utilization of learners' multimodal data is integral to the development and implementation of innovative pedagogical and curriculum strategies [Kim, 2019]. Specifically, collaboration learning has been promoted by utilizing digital technologies such as game-based learning, mobile learning, and simulations. Just placing individual learners in a group does not inherently signify evidence of collaborative learning facilitated through these technologies.

Growing interests has emerged regarding the integration of gaze behavior patterns to enhance idea improvements in collaborative learning. There is great evidence of successful interactions by the coordination of attention and gaze across a shared visual space [Brown-Schmidt *et al.*, 2005]. However, other multimodal learning data, gaze-based educational research is still in its early stages.

To address this challenge, this study aims to explore what are the main educational research trends of the use of gaze learning data through topic modeling and scientometric analysis. Our overall analysis was shaped by two research questions: (1) What are the main topics of gaze in educational research through topic modeling approach developed by our research team and scientometric analysis? and (2) What are the implications of the research findings on topic modeling and scientometric analysis as presented in this study?

## 2 Theoretical background

Educational research has indicated that multimodal learning data such as "linguistic, visual, audio, gestural, spatial and multimodal designs" [Kim, 2021], provide affordances, enabling embodied and more interactive opportunities for communication and meaning-making in fostering collaboration. Overall, the affordances of multimodal learning data empower both students and teachers to engage in more authentic and dynamic forms of learning across formal and informal contexts.

In this manner, gaze-based educational research also offers several benefits in understanding learning experiences and improving educational practices. These benefits of eye gaze data encompass (a) offering valuable insights into the level of learner engagement by tracking their visual attention towards targeted instructional materials or interventions, (b) tracking learners' specific learning process visually through monitoring their fixations and saccades, (c) identifying areas of difficulty or confusion employed by learners during learning tasks by measuring prolonged fixations and frequent regressions, and (d) enhancing learner interaction design and tools using technologies by examining user experience.

To quantitatively examine research trends in gaze educational research, we conducted a scientometric analysis emphasizing publication patterns, citation networks, and collaboration among educational researchers [Park *et al.*, 2023]. For such analysis, we adopt techniques such as topic modeling [Blei *et al.*, 2003a] and scientometric analysis to qualitatively and quantitatively illustrate the semantic shifts in educational papers related to gaze.

| Setting up search criteria | Database | Web of Science |
| --- | --- | --- |
| Initial retrieval | Search String | Gaze * (learning or teaching or education or instruction) |
| Fine-tuning retrieval | Retrieved abstract | 573 results |
| | Publication years | From 2008 to 2024 |
| | Document types | Article, Proceeding paper |
| | Languages | English |
| | Research areas | Education educational research |
| Final retrieval | Retrieved full papers | 463 results |

Table 1: **A summary of search criteria and procedure for data collection**.

| Indicators | Cited Reference |
| --- | --- |
| Time span (co-citation) | 2014-2024 |
| Nodes (cited references) | 683 |
| Edges (citations) | 1,999 |
| Density | 0.0086 |
| Modularity | 0.9073 |
| Mean Silhouette | 0.9683 |

Table 2: **A network summary of cited references**.

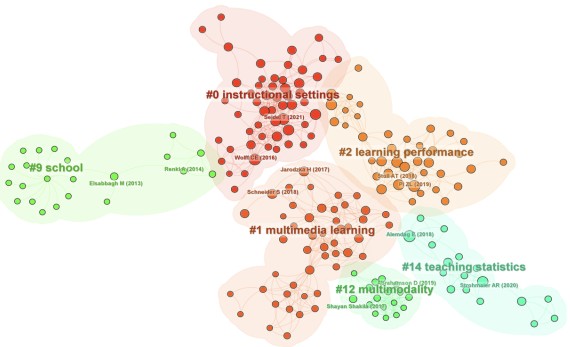

Figure 1: **Clusters for cited references**.

# 3 Document Analysis using Topic Model

In this section, we encapsulate the core principles of the topic model, highlighting two prevalent approaches: (1) semantic topic extraction across entire documents and (2) document clustering based on the identified topics. With $K$ topics denoted as $\beta_k \in \boldsymbol{\beta}, k = 1, ..., K$, the topic model allocates documents to one of these topics, constituting a clustering procedure based on the topics. This allocation can be deterministic or generative, achieved by specifying the topic distribution for each document as follows:

$$z_{dn} \sim p_{\theta_d}(z), \qquad (1)$$

In the generative process, the distribution $p_{\theta_d}(z)$ selects the index variable $z_{dn}$, representing the topic index $\beta_{z_{dn}}$ encompassing the word $w_{dn}$ within the $d$-th document. Typically, in a generative framework, the random variable $\boldsymbol{\theta}$ follows a $K$-dimensional Categorical distribution [Blei *et al.*, 2003a] with a Dirichlet prior $\alpha$, or a Product of Expert (PoE) [Srivastava and Sutton, 2017].

Each topic $\beta_k$ is characterized by a set of semantically coherent words $w_{kn} \in \beta_k, 1, ..., N_w$, or alternatively, by a generatively defined word distribution, as follows:

$$w_k \sim p_{\beta_k}(w). \qquad (2)$$

Similarly, $p_{\beta_k}(w)$ may adopt categorical-like distributions [Blei *et al.*, 2003a]. Classical probabilistic generative topic models [Blei *et al.*, 2003a; Srivastava and Sutton, 2017] interpret each document $d$ as a Bag-of-Words (BoW) $\mathbf{w}_d = w_{d1}, ..., w_{dn}$ and analyze the joint distribution $p(\boldsymbol{\theta}, \boldsymbol{\beta}|\mathbf{w}_d)$ from equations (1-2), employing approximated Bayesian inference methods [Casella and George, 1992; Wainwright *et al.*, 2008; Kingma and Welling, 2013].

When embedding is integrated into topic modeling frameworks [Dieng *et al.*, 2020; Meng *et al.*, 2022], certain branches of embedded topic models retain the word generation ability, thus incorporating word embedding into their probabilistic framework, as observed in ETM [Dieng *et al.*, 2020]. Non-generative embedded topic models, including recent PLM-based topic models [Sia *et al.*, 2020; Grootendorst, 2022a], directly extract topic embedding via distance-based clustering methods, circumventing complex Bayesian inference approximations.

# 4 Methodology

As illustrated in Table 1, the PRISMA procedure [Liberati *et al.*, 2009] was employed to identify relevant peer-reviewed or proceeding studies published in English since 2008, a period characterized by a substantial increase in scholarly output, particularly observed in the Web of Science database, renowned as one of the most popular databases in educational research. The process of selecting relevant educational studies on gaze was guided by keywords such as 'gaze' with 'learning' or 'teaching' or 'education' or 'instruction.'

## 4.1 Topic modeling

To embark on topic extraction and evaluation, we commence by preprocessing the input documents in accordance with the established conventions outlined in [Blei *et al.*, 2003a]. Upon varying the number of topics, our initial endeavor involves qualitatively visualizing the dominant words within each topic. Furthermore, to undertake a quantitative assessment of topic quality, we proceed to evaluate the model's efficacy concerning Topic Quality (TQ) and its ability to represent documents, in alignment with the standardized evaluation framework devised for topic models.

| ID | Size | Silhouette | Mean | Top 5 Terms (LLR) |
|----|------|-----------|------|-------------------|
| 0 | 54 | 0.963 | 2019 | Instructional settings Prompting Specific task instruction Mixed methods Preservice teacher education |
| 1 | 48 | 0.943 | 2016 | Multimedia learning Eye movement modeling examples Learning performance Teaching Eye gaze |
| 2 | 41 | 0.974 | 2019 | Learning performance Eye gaze Video lectures Facial expression Teacher research |
| 9 | 22 | 1.000 | 2012 | School Special education Social interaction Attention Teaching |
| 12 | 17 | 0.978 | 2017 | multimodality Mathematics Gesture Design Embodiment |
| 14 | 17 | 0.990 | 2019 | Teaching statistics Coordination dynamics Histogram Learning analytics Machine learning algorithm |

Table 3: **Cluster summary for cited references**.

The evaluation of TQ hinges upon two pivotal metrics: Topic Coherence (TC) and Topic Diversity (TD). TC is appraised through the utilization of cross-validated (CV) coherence, a metric devised to gauge the semantic coherence exhibited by the principal words encapsulated within each topic. The CV-coherence scores span a spectrum from 0 to 1, with higher values indicative of enhanced interpretability and semantic coherence. On the other hand, TD serves as a measure of word diversity, quantified by calculating the unique count of words amongst the top 25 words across all topics [Dieng et al., 2020]. TD scores range between 0 and 1, with elevated values signifying a more diverse array of words present.

## 4.2 Scientometric analysis

An analysis software, CiteSpace version 6.3.R3 [Chen, 2024], was adopted to analyze and visualize the citation patterns and the network of clusters of co-cited publications. In the network, a node indicates a cited reference or institutions where the publications were released. Two nodes are connected by an edge, indicating an occurrence of a citation. The clusters of the co-cited publications were visualized and entitled based on the titles, keywords, and abstracts of the co-cited publications [Chen, 2006]. Two indicators, modularity and silhouette, demonstrate how the network is structured. Specifically, modularity denotes the level of loosely distinctive division of the network into clusters. Silhouette indicates the homogeneity of the clusters in the network on average [Chen, 2024]. Further, a citation burst is created founded on a burst-detection algorithm [Kleinberg, 2002] "for detecting sharp increases of interest in a specialty," which is enabled and "identified based on such burst terms extracted from titles, abstracts, descriptors, and identifiers of bibliographic records" [Chen, 2006]. Thus, the publication with a burst is the emerging research with a spotlight in the field.

Figure 2: **Burstness for cited references**.

# 5 Findings

## 5.1 Scientometric analysis

**Cited reference analysis**

Tab. 2 indicated the density and modularity of the identified 6 clusters out of 132 clusters in the network of the cited references. The network has high modularity (0.9073) and silhouette (0.9683) values, indicating the homogeneity of the clusters.

Fig. 1 and Tab. 3 show the top six clusters identified through keyword analysis: "instructional settings" (54 references, the mean year of 2019, silhouette value of 0.963), "multimedia learning" (48 references, the mean year of 2016, silhouette value of 0.943), "learning performance" (41 references, the mean year of 2019, silhouette value of 0.974), "school" (22 references, the mean year of 2012, silhouette

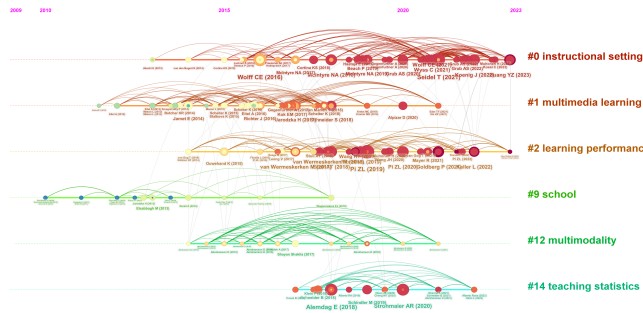

Figure 3: **Timeline for cited references**.

| Indicators | Institution |
|---|---|
| Time span (co-citation) | 2014-2024 |
| Nodes (institutions) | 238 |
| Edges (citations) | 221 |
| Density | 0.0075 |
| Modularity | 0.9073 |
| Mean Silhouette | 0.9683 |

Table 4: **A network summary of institutions**.

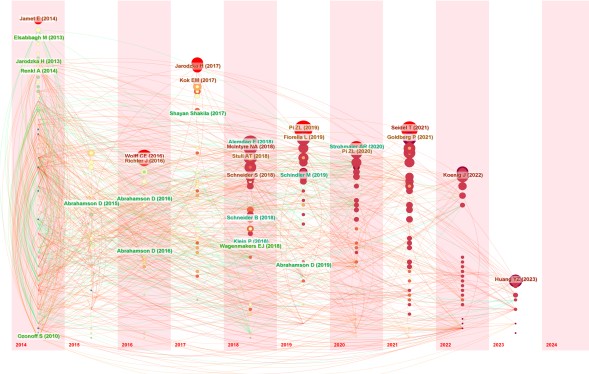

Figure 4: **Time zone for cited references**.

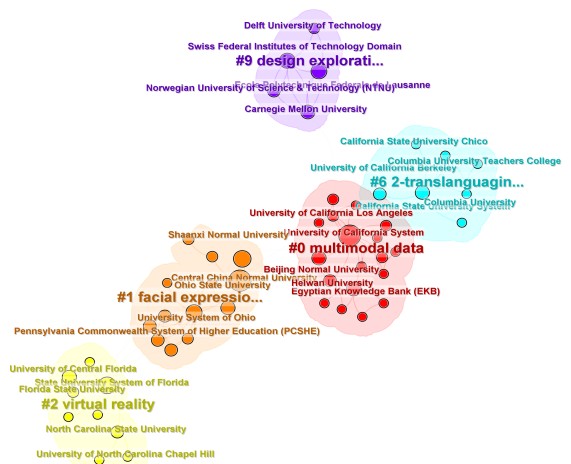

Figure 5: **Clusters for institutions**.

## Top 1 Institutions with the Strongest Citation Bursts

| Institutions | Year | Strength | Begin | End | 2014 - 2024 |
|---|---|---|---|---|---|
| Central China Normal University | 2019 | 3.59 | **2022** | 2024 | 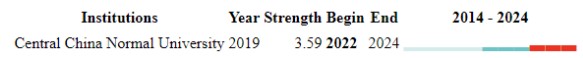 |

Figure 6: **Burstness for institutions**.

value of 1), "multimodality" (17 references, the mean year of 2017, silhouette value of 0.978), "teaching statistics" (17 references, the mean year of 2019, silhouette value of 0.99).

The examination of co-cited references revealed that publications have been increasingly cited over time. Fig. 2 shows the top 8 publications that were discovered through the citation burst detection. Since 2016, these studies have been cited abruptly.

In Fig. 3 and 4, the timeline and time zone provide a representation of the progression of cited references over time, offering insights into the development of research themes. The timeline and time zone visualizations demonstrate the significant evolution of gaze-related research from 2014 to 2024.

**Institution analysis**
Tab. 4 indicated the density and modularity of the identified 5 clusters out of 114 clusters in the network of the institutions. The network has high modularity (0.9073) and silhouette (0.9683) values the same as the network of cited references, indicating the homogeneity of the clusters.

Fig. 5 and Tab. 5 show the top five clusters identified through keyword analysis: "multimodal data" (18 references, the mean year of 2018, silhouette value of 0.926), "facial expression" (13 references, the mean year of 2020, silhouette value of 0.948), "virtual reality" (9 references, the mean year of 2017, silhouette value of 1), "2-translanguaging" (7 references, the mean year of 2019, silhouette value of 0.993), "design exploration" (5 references, the mean year of 2018, silhouette value of 1).

The examination of co-institution revealed that a publication has been cited explosively. Fig. 6 shows one publication from Central China Normal University that was discovered through citation burst detection. Since 2022, this study has been cited abruptly.

In Fig. 7 and 8, like cited references, the timeline and time zone represent the progression of institution publications over time, offering insights into developing research themes. The timeline and time zone visualizations demonstrate the significant evolution of gaze-related research from 2014 to 2024.

### 5.2 Topic modeling

**Quantitative evaluation.** We examine the performance of algorithms for topic modeling using LDA [Blei *et al.*, 2003b] and BERT [Grootendorst, 2022b]. Increasing the number of topics from ten to fifty at ten-topic intervals, we evaluate TC and TD subsequently by computing the mean of these met-

| ID | Size | Silhouette | Mean | Top 5 Terms (LLR) |
|---|---|---|---|---|
| 0 | 18 | 0.926 | 2018 | Multimodal data Students' performance Tangible user interfaces Embodied learning Eye tracking |
| 1 | 13 | 0.948 | 2020 | Facial expression Instructor-generated outlines Teacher preparation Pedagogy Instructor presence |
| 2 | 9 | 1.000 | 2017 | Virtual reality Engagement Simulation Scholarship of teaching and learning Visual attention |
| 6 | 7 | 0.993 | 2019 | 2-translanguaging 3-race Problem solving /decision making 1-early childhood Organic chemistry |
| 9 | 5 | 1.000 | 2018 | Design exploration Educational technology Vocational education and training Vet Its |

Table 5: **Cluster summary for institutions**.

rics. Results depicted in Fig. 9 reveal that LDA maintains a TC of 0.374, indicating the presence of semantically consistent topics to a certain extent. BERT, however, achieves a TC of 0.733, signaling a generation of topics with substantially greater consistency as compared to LDA. TD also displays a stark contrast, with LDA registering a value of 0.185 against BERT of 0.9275, highlighting a significantly wider array of topics from the latter.

The visual representation in Fig. 10 contrasts TQ for both LDA and BERT across varying topic counts. BERT consistently surpasses LDA in TQ across all topic quantities, indicating robustness in maintaining high-quality topics. This graph elucidates the responses of each algorithm to an array of topics, where BERT exhibits steadfast consistency, unlike the variability shown by LDA. These findings suggest a preference for BERT in applications of topic modeling, owing to its potential to enhance performance.

Fig. 11 shows the TC score for 40 individual topics within the scope of LDA and BERT. Each line on the graph reflects the TC score changes corresponding to topic indices, with the vertical axis representing the TC values and the horizon-

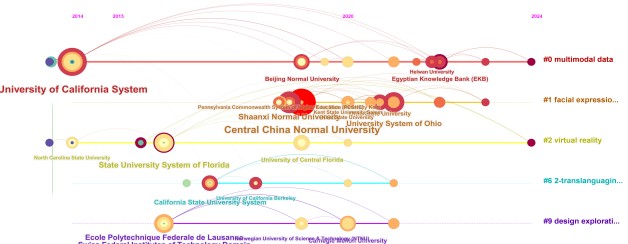

Figure 7: **Timeline for institutions**.

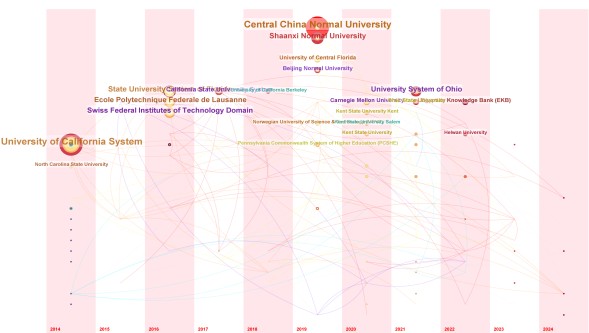

Figure 8: **Time zone for institutions**.

tal axis marking the topic numbers. LDA shows considerable fluctuation in TC scores, with some topics displaying notably lower consistency. Meanwhile, BERT consistently maintains higher TC values than those of LDA, signifying a superior level of topic consistency. Comparing the two models, BERT routinely achieves higher TC scores, implying the creation of topics with more robust and consistent semantic relations. The evidence suggests that BERT could outperform LDA in generating high-quality topics within the domain of topic modeling. The graph provides insights into the reaction of each topic modeling approach to the varied topics, underscoring the potential benefits of selecting BERT for enhanced topic modeling performance.

**Qualitative evaluation.** We selected the top ten words for each of the forty topics derived through each methodology and visualized these words in word clouds, as shown in Figs. 12 and 13.

The results based on LDA uncover standard topic structures within the educational domain. In contrast, the results obtained through BERT reveal a greater diversity of topics, such as inclusive education, multimodal learning, and educational psychology. The LDA-based topic modeling prominently features terms such as 'eye' and 'gaze,' reflecting a focus on eye tracking within educational research. Results based on BERT display distinct boundaries between topics, delineating various sub-areas of the educational field.

We also find that topics derived from BERT-based topic modeling show similarities with clustering outcomes from the scientometric analysis tool. Topic 1, comprising terms such as 'lecture' and 'instructor,' matches cluster ID 0 in CiteSpace, reflecting a concentration of research on teaching and instructors. Topic 0, featuring terms related to eye-tracking,

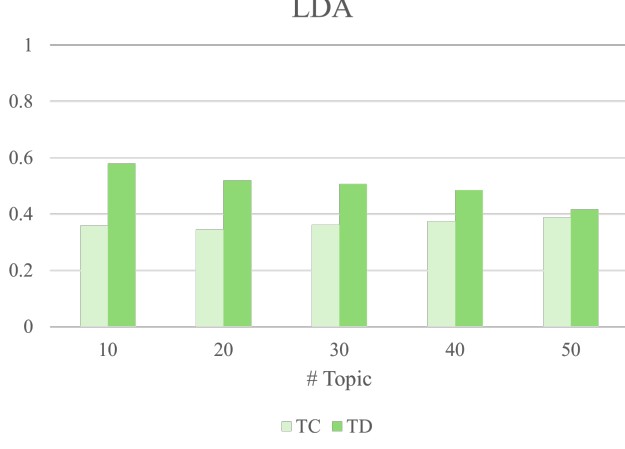

LDA

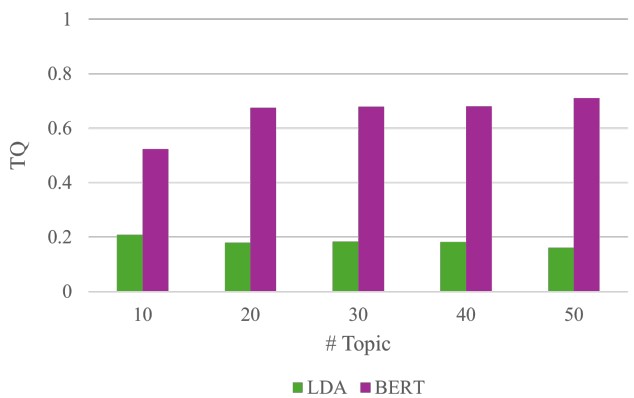

Figure 10: **Comparison of TQ**.

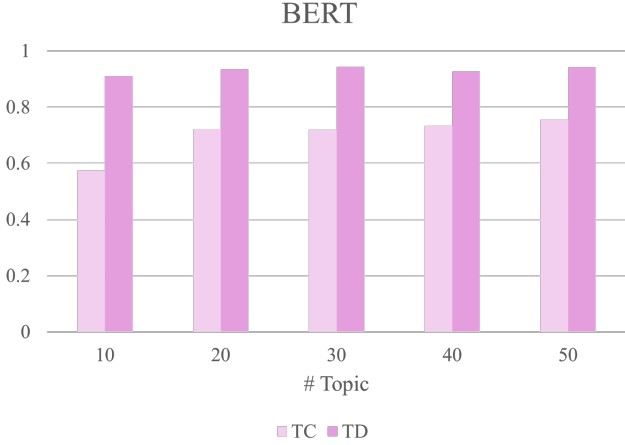

BERT

Figure 9: **Comparison of TC and TD**.

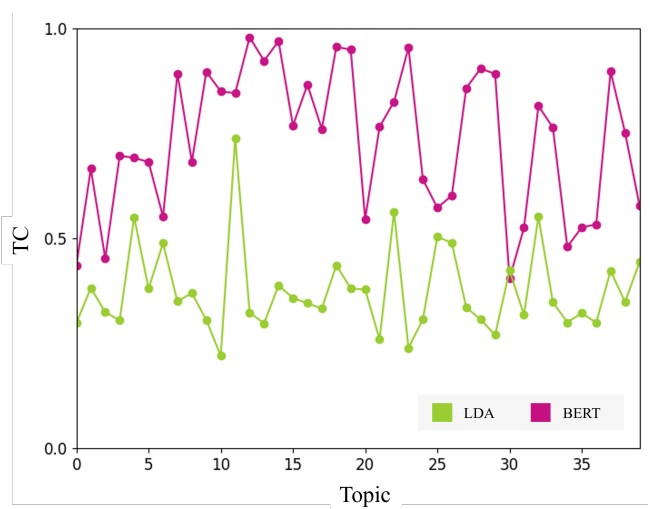

Figure 11: **TC scores for each of 40 topics**.

aligns with CiteSpace cluster ID 1. Topic 30, including terms like 'face' and 'video,' connects with cluster ID 2. Topic 3, concerning special education and attention, shows similarity to cluster ID 9, and Topic 17, encompassing mathematical and algorithmic modalities, corresponds with cluster ID 12.

When comparing with the institution-based CiteSpace analysis results, we observe that Topic 40, with terms such as 'facial' and 'expression,' could link to cluster ID 1, focusing on facial expression recognition research. Topic 39, containing terms like 'design' and 'vocational', shares similarities with cluster ID 9, which includes research in vocational education. The analysis confirms a meaningful correlation between the topics derived from the modeling and the clustering results from CiteSpace, suggesting that BERT effectively identifies and categorizes various topics in gaze-based educational research, thereby performing a complementary role to scientometric analysis.

BERT demonstrates its potential to contribute significantly to research analysis by providing meaningful topics, even in the absence of high semantical information. For example, topics generated from BERT show direct correspondence with CiteSpace clusters, affirming their effectiveness in detecting the interconnections among gaze-based educational

topics and the evolving trends in academic networks. Comparative analyses like these aid in discovering principal topics and their impact on educational research, offering substantial insights into the educational dynamics and learning processes facilitated by human gaze data.

## 6 Discussion

By employing topic modeling methods based on LDA and BERT, alongside scientometric analysis, we identified topics and trends within gaze-based educational research published in the Web of Science from 2008 to 2024. Our analysis revealed a meaningful alignment between the topics uncovered through BERT-based topic modeling and the clusters obtained from scientometric analysis. Specifically, BERT effectively identified and categorized a diverse range of consistent topics in gaze-based educational research compared to LDA.

In the LDA, terms associated with 'eye' and 'gaze' prominently emerged. In contrast, BERT displayed clearly defined boundaries between topics, effectively revealing various sub-areas within the educational field. Each topic identified through BERT aligned with the clustering outcomes obtained

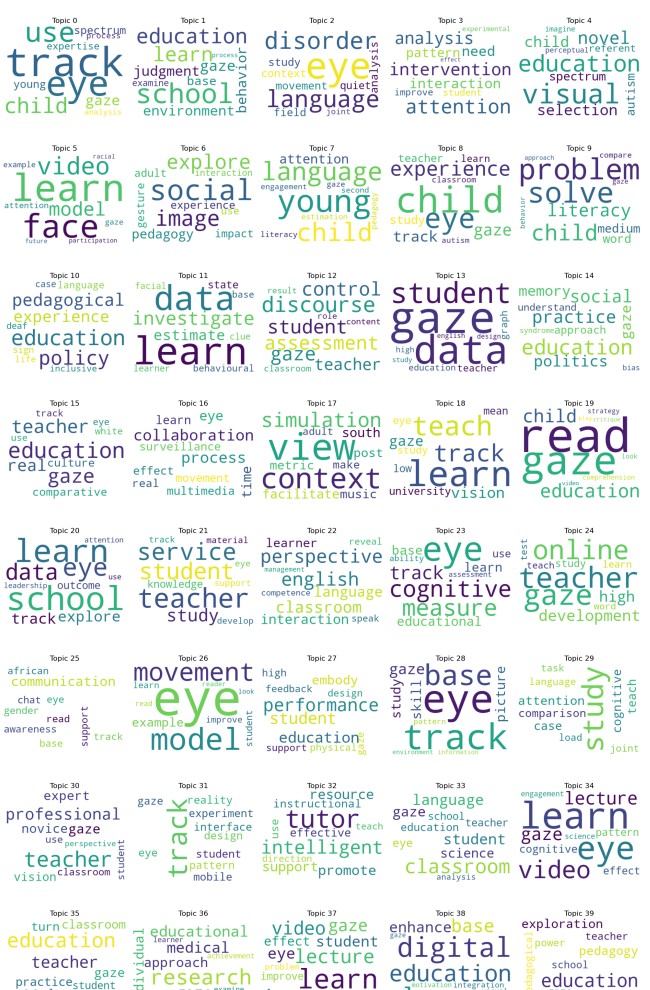

Figure 12: **Word clouds of LDA-based topic modeling**.

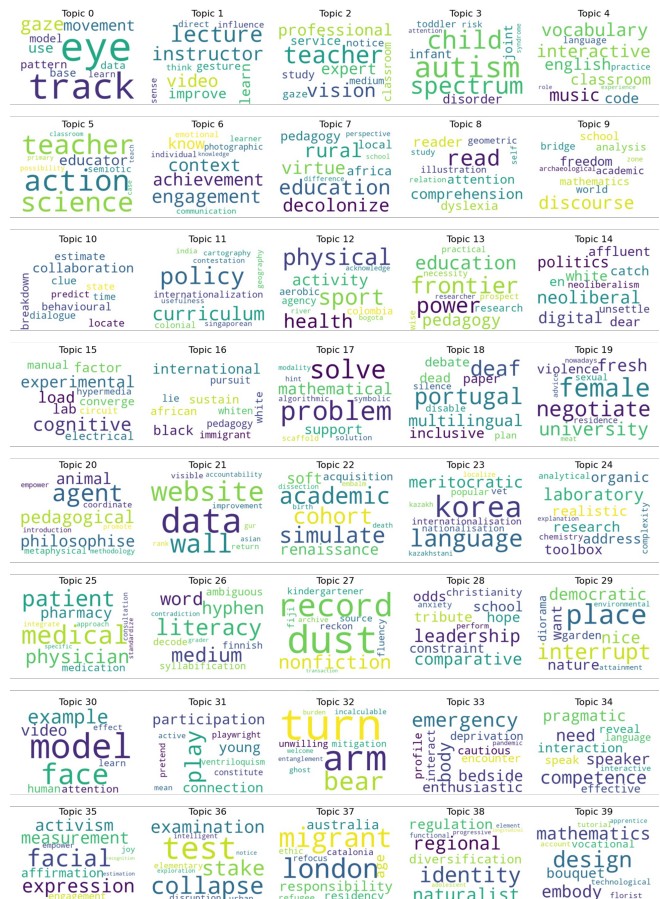

Figure 13: **Word clouds of BERT-based topic modeling**.

through scientometric analysis. Significant topics such as inclusive education, multimodal learning, and educational psychology surfaced in both methods. These findings demonstrated the capability of BERT to effectively distinguish key topics, even without the high semantic information typical of scientometric analysis, thereby validating its crucial role in literature analysis.

Our findings highlighted the interconnectedness between gaze-based educational research and computer vision and underscored the potential for further collaboration and development. This study inspired prominent interdisciplinary approaches to more culturally inclusive and interactive visual learning interfaces across diverse educational settings and contexts.

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
