# OpenReview forum: "Educational Research Trends of the Use of Gaze Learning Data through Topic Modeling and Scientometric Analysis"
_ijcai.org/IJCAI/2024/Workshop/AI4Research — AI4Research 2024_

### Official Review · Reviewer_zMLC · 2024-06-01
**Relevant topic, appropriate methods, but weakness in scope, rationale, and discussion/implications**

**Rating:** 6
**Confidence:** 5

**Review:**

The author used topic modeling and scientometric (Citespace) to analyze the research trend on the use of gaze in education. This paper's topic is relevant to the conference, and the methods used are appropriate. The main weaknesses are the scope, rationale, and discussion/implication.

1. The scope of the study is unclear and ill-defined. I don't really understand what "gaze-based educational research" entails. It seems like the authors are focusing on using gaze in collaborative learning. However, the field of education is much broader. The eye-tracking technique, including analyzing gaze data, has been used in many fields, including educational psychology, applied linguistics, science education, educational technology, and many more. It seems like the authors lack a clear understanding of the field of education, which led to the vagueness in the scope and position of this paper.

2. The rationale for studying this topic is not strong enough. Why do you want to analyze gaze in education? The authors did provide some benefits of gaze data in the theoretical background section. Still, it is more like a justification after the authors chose to study this topic rather than a rationale acknowledging the importance and telling the readers why this topic is important. For me, the choice of gaze is fairly arbitrary, and some questions are unaddressed. For instance, why didn't the authors choose to use "eye-tracking", which is a broader term and covers more paper? Choosing "gaze" instead of "eye-tracking" is like choosing "epoch" instead of "machine learning". It's not like you cannot focus on this, but why?

3. The discussion and implications are fairly vague and didn't really provide useful information other than summarizing the study. This is probably related to the lack of in-depth understanding of the field of education. Some additional information that will be useful include: "What are some of the areas that are emerging and that can benefit from the use of gaze data?" and "What are the research gaps suggested by the data?"

---

### Official Review · Reviewer_fZaf · 2024-06-02
**Analysis of main topics of gaze in educational research**

**Rating:** 7
**Confidence:** 4

**Review:**

This paper utilizes document and scientometric analysis to identify the main educational research trends in gaze learning. It first identifies relevant peer-reviewed studies published in English since 2008 using specific keywords. Then, it evaluates the topic quality with two metrics: TC and TD, using topic modeling models LDA and BERT. TC represents topic semantic coherence, while TD is measured by word diversity. Subsequently, it adapts analysis software CiteSpace to apply scientometric analysis, which includes analyzing and visualizing citation clusters and patterns. Finally, the paper summarizes the findings with topic keywords and research institution analysis.

Strengths:
1. The paper is well written, and the theoretical background is very helpful for an audience from different backgrounds.
2. The reference paper search is very thorough and detailed.

Weaknesses:
1. Some figures are not very readable, such as Figures 3, 4, 7, and 8.
2. Some keywords are not meaningful, like 'Its'. Is there a way to clean the data first and then process it?
3. The author claims that the findings demonstrate the capability of BERT to effectively distinguish key topics, but there is no ground truth to evaluate these findings. There may be better models, such as fine-tuned BERT with education data or new LLMs.

---

### Decision · Program_Chairs · 2024-06-03

Accept